# DYN-CONVNET: HYBRID TIME–FREQUENCY LEARNING FOR TIME SERIES ANOMALY DETECTION

## ABSTRACT

Time series anomaly detection (TSAD) is of critical importance in applications such as industry, finance, and healthcare, yet it remains challenging due to complex temporal dependencies, non-stationarity, and the scarcity of anomalous samples. Existing methods typically operate in a single domain—either time or frequency—limiting their ability to capture diverse anomalies across both global and local patterns. To address this, we propose Dyn-ConvNet, a novel hybrid deep learning architecture that systematically integrates time- and frequency-domain representations by combining the Fast Fourier Transform (FFT) for long-range periodic feature extraction with the Wavelet Transform for local and transient anomaly detection. These complementary features are fused through a deep convolutional backbone enhanced with gating mechanisms and residual connections, enabling adaptive learning and robust detection across different scales and anomaly types. Experiments on five popular multivariate time series benchmark datasets show that Dyn-ConvNet outperforms state-of-the-art methods, with even larger gains in complex anomaly scenarios, demonstrating the effectiveness of multi-domain feature integration in enhancing both the performance and generalization capability of multivariate time series anomaly detection.

## 1 INTRODUCTION

Time series anomaly detection (TSAD) is a critical task in diverse domains, from industrial control systems and financial risk management to medical diagnostics and IT operations(Wu et al., 2021; Franceschi et al., 2019; Friedman, 1962; Xu et al., 2021). It aims to identify data points or subsequences that significantly deviate from normal patterns. Unlike anomalies in static data, time series anomalies are complex, manifesting as point anomalies (single abnormal data points), contextual anomalies (data points that are abnormal only within a specific temporal context), or collective anomalies (subsequences that are abnormal as a whole). These anomalies are challenging to detect due to complex temporal dependencies, non-stationarity, and inherent noise. Furthermore, their rare, diverse, and context-dependent nature makes robust model learning and generalization particularly difficult.

Traditional statistical methods, such as autoregressive models(Box et al., 2015), are interpretable but often fail to model non-linear dependencies or adapt to evolving data distributions. While deep learning models—including recurrent neural networks (RNNs), convolutional neural networks (CNNs)(Lai et al., 2018), and attention-based Transformers(Wu et al., 2021), have shown notable progress, most existing architectures are constrained to a single domain of representation. Time-domain models excel at capturing local sequential patterns but struggle with long-range periodicity. In contrast, frequency-domain approaches like the Fast Fourier Transform (FFT) effectively uncover global trends and seasonality(Papoulis & Saunders, 1989), but they are ill-suited for detecting transient, non-periodic variations and require careful handling of cutoff frequencies.

Single-domain models therefore have clear limitations: time-domain methods can detect local abrupt spikes but are insensitive to global periodic deviations, whereas frequency-domain methods reveal periodic changes but often overlook short-lived anomalies(Oppenheim, 1999). To illustrate this, Figure 1 presents typical examples in both time and frequency domains. Some anomalies are subtle periodic deviations best seen in the frequency domain, while others are sudden spikes best detected through local time-domain analysis. Existing architectures often lack a robust mechanism for

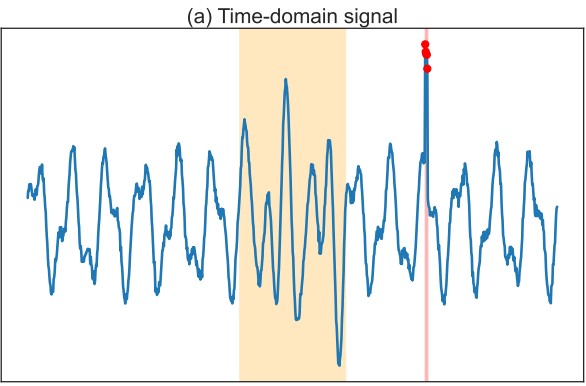
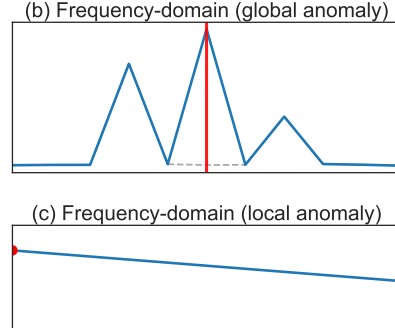

Figure 1: Limitations of single-domain time series anomaly detection. (a) Time-domain signal with local spikes (red) and a global periodic anomaly (orange). (b) Frequency spectrum of the global anomaly segment, showing a clear 2.5 Hz peak (red line) absent in the baseline (gray dashed). (c) Frequency spectrum of the local spike segment, where frequency components (red dot) are weak, illustrating that short-lived anomalies are not well captured in the frequency domain.

combining these complementary perspectives, causing single-domain models to inevitably overlook certain anomalies. To overcome these limitations, we propose Dyn-ConvNet, a novel hybrid deep learning model for robust time series anomaly detection. Dyn-ConvNet systematically integrates complementary information from both the time and frequency domains. We leverage FFT to capture global periodic patterns and Wavelet Transform to pinpoint localized, transient anomalies. These features are then fused through a deep convolutional network enhanced with gating mechanisms and residual connections, enabling adaptive learning and robust operation across diverse scenarios, from subtle deviations to sudden, pronounced anomalies.

Our main contributions are summarized as follows:

- We propose Dyn-ConvNet, a novel hybrid architecture that, for the first time, systematically and adaptively integrates global periodic patterns captured by FFT with localized transient features identified by Wavelet Transform.

- We design an adaptive feature fusion mechanism with gating and residual structures, significantly enhancing the model's robustness against heterogeneous anomaly patterns.

- Through extensive experiments on multiple real-world benchmark datasets (SMD (Su et al., 2019), MSL (Hundman et al., 2018), SMAP (Hundman et al., 2018), SWaT (Mathur & Tippenhauer, 2016), PSM (Abdulaal et al., 2021)), we demonstrate that Dyn-ConvNet significantly outperforms state-of-the-art methods, achieving an average performance improvement of 1.46%.

## 2 RELATED WORK

The landscape of time series anomaly detection (TSAD) has evolved significantly, beginning with traditional statistical models that laid the foundational principles. Early approaches, such as autoregressive (AR) models(Box et al., 2015), exponential smoothing(Holt, 2004), and hypothesis testing(Barnett et al., 1994), offered interpretability and explicit modeling of normal behavior. However, they impose strong assumptions on data stationarity and anomaly distributions, making them rigid and ill-suited for the non-stationary, high-noise, and dynamically changing environments of real-world data. Moving beyond these constraints, classical machine learning techniques like support vector machines (SVM)(Schölkopf et al., 2001), isolation forests(Liu et al., 2008), and Gaussian mixture models (GMM) were applied to TSAD(Bishop & Nasrabadi, 2006). While these methods can capture nonlinear relationships, they often rely on handcrafted features and have a limited ability to model long-term temporal dependencies.

In recent years, deep learning methods have achieved remarkable performance in TSAD by moving beyond handcrafted features. Recurrent neural networks (RNNs) and their variants (LSTMs, GRUs) excel at capturing sequential dependencies(Lai et al., 2018; Hochreiter & Schmidhuber, 1997; Franceschi et al., 2019; Gu et al., 2022), making them effective for detecting point and contextual

anomalies. Similarly, convolutional neural networks (CNNs) leverage convolutional kernels to extract local patterns, enabling high sensitivity to sudden spikes and local perturbations(Wang et al., 2017). More recently, attention-based models, such as Transformers(Zhou et al., 2021; Liu et al., 2021; Wu et al., 2021; Zhou et al., 2022; Vaswani et al., 2017b), have proven effective in modeling long-range dependencies and global patterns. Despite their strengths in modeling time-domain features, these methods often struggle to detect periodic anomalies or global trend shifts when relying on time-domain information alone.

To address this limitation, frequency-domain approaches analyze periodicity and trends by transforming time series into their frequency components. Methods like the Fast Fourier Transform (FFT) efficiently capture global periodic patterns but perform poorly in detecting transient, non-periodic anomalies(Oppenheim, 1999; Choi et al., 2021). Wavelet transforms provide a joint time-frequency analysis but remain limited in their sensitivity to local patterns when used in isolation(Mallat, 2002). While some studies have proposed hybrid approaches that fuse features to exploit these complementary strengths, existing models are often shallow or rely on simple feature concatenation. This simple fusion lacks an adaptive mechanism to intelligently weigh information from different domains, thus failing to achieve deep representation learning and limiting their robustness across heterogeneous datasets and anomaly patterns.

To overcome these challenges, we propose Dyn-ConvNet, a novel hybrid model that not only fuses time- and frequency-domain features but does so through a dynamic, deep learning architecture. This systematic integration enables adaptive learning and robust detection across a wide range of anomalies, from subtle periodic deviations to abrupt spikes—a capability lacking in prior work.

## 3 DYN-CONVNET

### 3.1 PROBLEM DEFINITION

We address the task of time series anomaly detection (TSAD), where the goal is to identify irregular patterns that deviate from expected behavior. Given an input sequence $X = \{x_1, x_2, \ldots, x_T\}$, the model reconstructs an approximation $\hat{X} = g(X)$. An anomaly score is then computed for each time step as

$$s_t = \|x_t - \hat{x}_t\|_2^2, \tag{1}$$

where higher scores indicate stronger deviations.The challenge arises from the dual nature of anomalies: Local irregularities (e.g., spikes, bursts) that require fine-grained temporal modeling(Lai et al., 2018).Global periodic deviations (e.g., shifts in frequency or amplitude) that require spectral analysis(Wu et al., 2021).A model optimized for one type often fails on the other. Dyn-ConvNet is designed to unify both perspectives through a joint time–frequency architecture.

### 3.2 OVERALL FRAMEWORK

Figure 2 illustrates the overall architecture of Dyn-ConvNet. Given an input sequence $X \in \mathbb{R}^{B \times 1 \times L}$, the model processes it via two parallel feature streams. The Fourier feature stream utilizes FFT to analyze the signal's spectral content and identify dominant periodicities. The identified periodic components are then used to generate weighted cosinusoidal and sinusoidal features. These, along with the original input, are processed by a $1 \times 1$ convolution and layer normalization to produce $H_{\text{fourier}}$.

The wavelet feature stream employs multi-scale convolutions (kernel sizes 3 and 5) to extract local patterns, producing $H_{\text{wavelet}}$. A gating unit, informed by the Fourier features, generates weights to modulate $H_{\text{wavelet}}$, resulting in $H_{\text{gated}}^{\text{wavelet}}$.

The fusion module concatenates $H_{\text{fourier}}$ and $H_{\text{gated}}^{\text{wavelet}}$, followed by a $1 \times 1$ convolution and layer normalization to yield $H_{\text{fused}}$. A core convolution block with residual connections refines $H_{\text{fused}}$, and a final projection layer outputs the reconstructed sequence $\hat{X} \in \mathbb{R}^{B \times L}$. This parallel and gated design enables joint learning of global periodic and local multi-scale features, enhancing anomaly detection performance.

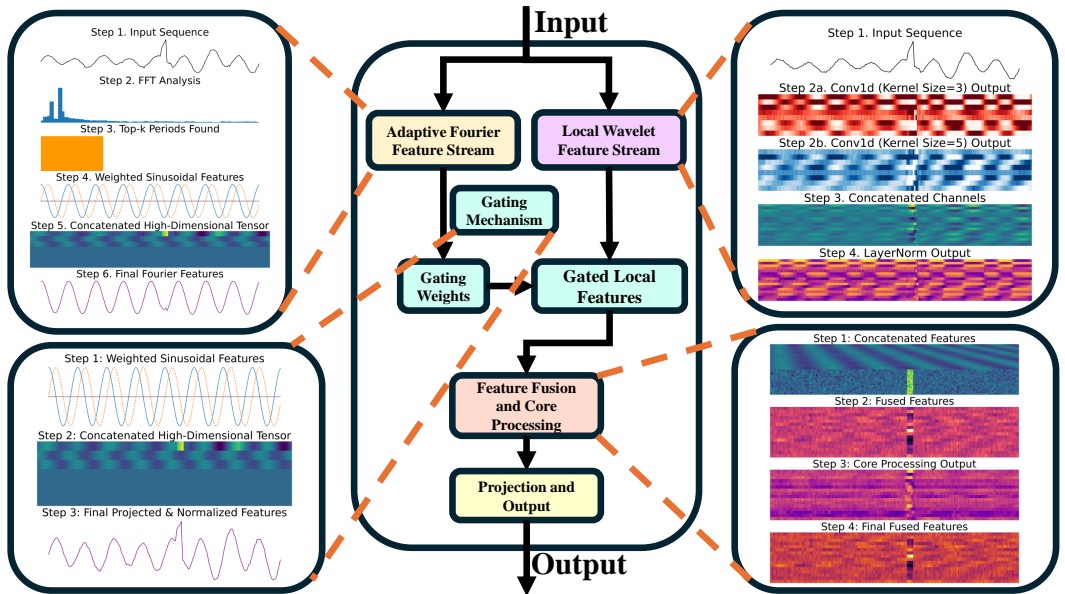

Figure 2: Dyn-ConvNet architecture. The model integrates a Fourier branch for global periodicity and a wavelet branch for local multi-scale features, with a novel Fourier-guided gating mechanism that adaptively modulates the wavelet stream before fusion for improved anomaly detection.

### 3.3 MODEL ARCHITECTURE DETAILS

#### 3.3.1 FOURIER FEATURE STREAM

Time-series data often contain global periodic patterns that reveal underlying regularities, but these patterns can be obscured by noise or non-stationarity. To address this, Dyn-ConvNet introduces the Fourier Feature Stream, which explicitly transforms the input sequence into the frequency domain to identify dominant periodicities. By applying FFT, the model captures the global spectral structure of the sequence:

$$X_f = \text{FFT}(X). \tag{2}$$

A learnable convolution kernel $W_f$ is then applied to selectively emphasise informative frequency components:

$$H^{\text{freq}} = \text{Conv}(X_f; W_f). \tag{3}$$

This can be viewed as an adaptive spectral filter, allowing the model to go beyond raw FFT amplitudes. Finally, inverse FFT maps the filtered features back to the time domain:

$$H^{\text{freq}}_{\text{time}} = \text{iFFT}(H^{\text{freq}}), \tag{4}$$

Ensuring compatibility with downstream modules. This stream is designed to provide robust global context for anomaly detection, especially for periodic anomalies.

#### 3.3.2 WAVELET FEATURE STREAM

Local anomalies tend to be subtle and context-dependent. Fixed convolution kernels are often insufficient to capture such variability. Dyn-ConvNet therefore introduces a Wavelet Feature Stream equipped with adaptive dynamic convolution, enabling the model to extract fine-grained temporal patterns in a context-sensitive way(Chen et al., 2020). For each local window $X_{i:i+k}$, a kernel generator $f_\theta$ produces an input-dependent kernel:

$$W_i = f_\theta(X_{i:i+k}), \qquad H^{\text{time}}_i = \text{Conv}(X_{i:i+k}; W_i). \tag{5}$$

This mechanism allows the model to adjust its receptive field dynamically, effectively focusing on the most informative local patterns.

### 3.3.3 GATING AND FUSION MODULE

Global periodicity and local patterns offer complementary insights, but their relative importance can vary across time and context. Instead of naively combining them, Dyn-ConvNet employs a Gating and Fusion Module that adaptively integrates the two streams. Specifically, the gate is computed as:

$$G = \sigma\big(W_g[H^{\text{time}}; H^{\text{freq}}_{\text{time}}] + b_g\big) \tag{6}$$

where $\sigma$ denotes the sigmoid function and $W_g, b_g$ are learnable parameters. This gate dynamically balances contributions from both streams. The fused representation is then obtained by:

$$H^{\text{fused}} = G \odot H^{\text{time}} + (1 - G) \odot H^{\text{freq}}_{\text{time}}. \tag{7}$$

This mechanism ensures that Dyn-ConvNet emphasizes global periodicity when relevant, while prioritizing local details for abrupt anomalies.

### 3.3.4 CORE CONVOLUTION BLOCK AND PROJECTION

The fused representation $H^{\text{fused}}$ is further refined through a residual convolution block:

$$H^{(l+1)} = \text{Conv}\big(\text{ReLU}\big(\text{Conv}(H^{(l)})\big)\big) + H^{(l)}, \qquad H^{(0)} = H^{\text{fused}}. \tag{8}$$

This block enhances feature interactions while preserving important information. Finally, a projection layer maps features back to the original dimension:

$$\hat{X} = \text{Proj}\big(H^{(L)}\big), \tag{9}$$

and anomaly scores are computed as reconstruction errors.

## 4 EXPERIMENTS

### 4.1 DATASETS

We evaluated our Dyn-ConvNet model on five widely-used public benchmark datasets for multivariate time series anomaly detection: SMD (Su et al., 2019), MSL (Hundman et al., 2018), SMAP (Hundman et al., 2018), SWaT (Mathur & Tippenhauer, 2016), PSM (Abdulaal et al., 2021). These datasets provide a diverse and comprehensive testbed, encompassing monitoring data from various domains, including water treatment systems, server information, and spacecraft telemetry. Each dataset presents unique challenges, from series with strong periodic patterns to those characterized by abrupt, transient anomalies. A detailed description of each dataset can be found in Appendix A.

### 4.2 COMPARED MODELS

To evaluate Dyn-ConvNet, we compare it against a diverse set of state-of-the-art models in time series analysis. Our selection includes powerful Transformer-based models like Informer(Zhou et al., 2021), Autoformer(Wu et al., 2021), and TimesNet(Wu et al., 2022), which excel at capturing long-term dependencies. We also benchmark against efficient variants like Reformer(Kitaev et al., 2020) and FEDformer(Zhou et al., 2022), as well as simpler, yet effective, linear models such as DLinear(Zeng et al., 2023) and LightTS(Zhang et al., 2022). Finally, we include KAN-AD(Zhou et al., 2024), a direct competitor designed for anomaly detection, to highlight our model's specific advantages. This comprehensive comparison allows us to validate the effectiveness and robustness of our approach.

### 4.3 EXPERIMENTAL SETTINGS

Our experiments used a sliding window of 100 to segment the time series data. All models were trained with a batch size of 128 using the Adam optimizer at a learning rate of 0.0001. We evaluated performance using Precision, Recall, and F-score.

Table 1: Anomaly detection task. The P, R and F1 represent the precision, recall and F1-score (%) respectively. F1-score is the harmonic mean of precision and recall. A higher value of P, R and F1 indicates a better performance.

| Datasets | SMD | | | MSL | | | SMAP | | | SWaT | | | PSM | | | Avg F1 |
|---|---|---|---|---|---|---|---|---|---|---|---|---|---|---|---|---|
| Metrics | P | R | F1 | P | R | F1 | P | R | F1 | P | R | F1 | P | R | F1 | (%) |
| Transformer | 83.58 | 76.13 | 79.56 | 71.57 | 87.37 | 78.68 | 89.37 | 57.12 | 69.70 | 68.84 | 96.53 | 80.37 | 62.75 | 96.56 | 76.07 | 76.88 |
| Reformer | 82.58 | 69.24 | 75.32 | 85.51 | 83.31 | 84.40 | 90.91 | 57.44 | 70.40 | 72.50 | 96.53 | 82.80 | 59.93 | 95.38 | 73.61 | 77.31 |
| Informer | 86.60 | 77.23 | 81.65 | 81.77 | 86.48 | 84.06 | 90.11 | 57.13 | 69.92 | 70.29 | 96.75 | 81.43 | 64.27 | 96.33 | 77.10 | 78.83 |
| Pyraformer | 85.61 | 80.61 | 83.04 | 83.81 | 85.93 | 84.86 | 92.54 | 57.71 | 71.09 | 87.92 | 96.00 | 91.78 | 71.67 | 96.02 | 82.08 | 82.57 |
| Autoformer | 88.06 | 82.35 | 85.11 | 77.27 | 80.92 | 79.05 | 90.40 | 58.62 | 71.12 | 89.85 | 95.81 | 92.74 | 99.08 | 88.15 | 93.29 | 84.26 |
| DLinear | 83.62 | 71.52 | 77.10 | 84.34 | 85.42 | 84.88 | 92.32 | 55.41 | 69.26 | 80.91 | 95.30 | 87.52 | 98.28 | 89.26 | 93.55 | 82.46 |
| ETSformer | 87.44 | 79.23 | 83.13 | 85.13 | 84.93 | _85.03_ | 92.25 | 55.75 | 69.50 | 90.02 | 80.36 | 84.91 | 99.31 | 85.28 | 91.76 | 82.87 |
| LightTS | 87.10 | 78.42 | 82.53 | 82.40 | 75.78 | 78.95 | 92.58 | 55.27 | 69.21 | 91.98 | 94.72 | _93.33_ | 98.37 | 95.97 | 97.15 | 84.23 |
| FEDformer | 87.95 | 82.39 | 85.08 | 77.14 | 80.07 | 78.57 | 90.47 | 58.10 | 70.76 | 90.17 | 96.42 | 93.19 | 97.31 | 97.16 | 97.23 | 84.97 |
| TimesNet | 88.66 | 83.14 | _85.81_ | 83.92 | 86.42 | **85.15** | 92.52 | 58.29 | 71.52 | 86.76 | 97.32 | 91.74 | 98.19 | 96.76 | _97.35_ | 86.31 |
| KAN-AD | 87.13 | 84.21 | 84.08 | 88.98 | 71.13 | 79.06 | 93.21 | 81.27 | **86.83** | 92.85 | 93.50 | 93.17 | 99.35 | 92.73 | 95.93 | _87.81_ |
| Dyn-ConvNet | 89.40 | 88.48 | **88.94** | 89.67 | 75.40 | 81.92 | 91.92 | 72.12 | _80.55_ | 93.82 | 95.54 | **94.67** | 98.48 | 96.48 | **97.47** | **89.27** |

## 4.4 OVERALL PERFORMANCE AND ANOMALY TYPE ANALYSIS

As shown in Table 1, Dyn-ConvNet obtains the highest average F1 across the five benchmarks. It obtains an average F1-score of 89.27%, outperforming the next best model (KAN-AD(Zhou et al., 2024)) by over 1.46%. This demonstrates the superior capability of our hybrid time-frequency approach.

Dyn-ConvNet's strong performance can be attributed to its specialized architecture tailored to different anomaly types. On datasets dominated by periodic patterns, such as SMD(Su et al., 2019) and MSL(Hundman et al., 2018), the model leverages FFT features to effectively capture normal temporal dynamics, yielding F1-scores of 88.94% and 81.92%, respectively. For datasets characterized by abrupt, transient anomalies, like SWaT(Mathur & Tippenhauer, 2016) and PSM(Abdulaal et al., 2021), the inclusion of wavelet analysis enables top scores of 94.67% and 97.47%, reflecting high sensitivity to sudden deviations. Notably, on the challenging SMAP(Hundman et al., 2018) dataset, which contains a diverse mix of anomaly types, Dyn-ConvNet achieves a competitive F1-score of 80.55%, outperforming most baselines. This highlights the robustness of our adaptive fusion mechanism, which dynamically balances contributions from different feature streams to address complex anomaly distributions.

## 4.5 ABLATION AND INTERPRETABILITY STUDIES

To systematically evaluate the contribution of each key component in our proposed Dyn-ConvNet model, we conducted a comprehensive ablation study. Our experimental design is structured to progressively remove core modules, allowing us to isolate and measure their impact on model performance. We focus primarily on the F1-score, as it provides a balanced evaluation of the model's precision and recall, a critical metric for anomaly detection.

### 4.5.1 EXPERIMENTAL DESIGN

**Full Model** This is our complete architecture, including all core components, and serves as the baseline for performance evaluation.

**No-Gating** This model removes the dynamic gating unit, using simple feature concatenation to fuse the Fourier and Wavelet feature streams instead. This variant helps us directly evaluate the contribution of the gating mechanism itself.

**No-FFT** This variant removes the entire Fourier feature stream and the gating unit. It relies solely on the wavelet feature stream for anomaly detection, allowing us to evaluate the independent contribution of the local wavelet features.

Table 2: Ablation study of different model versions across multiple datasets. All results are reported using the F1-Score metric.

| Model Version | F1-Score | | | | |
|---|---|---|---|---|---|
| | SMD | MSL | SMAP | SWaT | PSM |
| Full Model | **0.8894** | **0.8192** | 0.8055 | **0.9467** | **0.9747** |
| - No Gating | 0.8822 | 0.8044 | 0.6869 | 0.9325 | 0.9742 |
| - No FFT | 0.8672 | 0.8096 | **0.8358** | 0.9384 | 0.9727 |
| - No Wavelet | 0.8765 | 0.8083 | 0.6848 | 0.9182 | 0.9745 |
| - No CoreConv | 0.8822 | 0.8138 | 0.7242 | 0.9095 | 0.9738 |

**No-Wavelet**   This variant removes the entire wavelet feature stream and the gating unit. It relies exclusively on the Fourier feature stream, allowing us to evaluate the independent contribution of the global periodic features.

**No-CoreConv**   This model removes the two-layer convolutional block that refines the fused features. This helps us assess the role of these layers in feature refinement and abstraction.

### 4.5.2   RESULTS AND ANALYSIS

The results of our ablation study are summarized in Table 2.

**Gating Mechanism**   By comparing the Full Model to the No-Gating variant, we validate the effectiveness of our dynamic gating mechanism. Removing the gating unit consistently leads to a performance drop across all datasets, demonstrating that our adaptive fusion approach is superior to simple feature concatenation. For instance, on the SMAP dataset, this removal results in a significant F1-score drop from 0.8055 to 0.6869.

**Wavelet Features**   To evaluate the independent contribution of wavelet features, we compare the No-Gating model to the No-Wavelet model. The results show that the role of wavelet features is highly dataset-dependent. On the SWaT dataset, their removal causes a significant performance degradation, with the F1-score dropping from 0.9325 to 0.9182. However, on other datasets like SMAP, MSL, and PSM, removing the wavelet features results in only minor changes or even a slight performance increase, suggesting that their contribution is not universally critical.

**FFT Features**   Similarly, to evaluate the independent contribution of Fourier features, we compare the No-Gating model to the No-FFT model. The impact of FFT features also varies significantly by dataset:On the SMAP dataset, removing the FFT features leads to a notable performance gain, with the F1-score increasing from 0.6869 to 0.8358. This suggests that the periodic features captured by FFT may be irrelevant or introduce noise for this dataset.On the SWaT and MSL datasets, removing the FFT features also results in a slight performance increase.Conversely, on the SMD and PSM datasets, removing the FFT features leads to a slight performance decrease, indicating their positive contribution.

**Core Convolution Block**   The ablation of the core convolutional block (No-CoreConv) demonstrates its necessity for the model's final performance. Comparing the Full Model to the No-CoreConv variant shows a consistent performance drop across all datasets when this block is removed. This confirms that the core convolutional block plays a crucial role in further refining and abstracting the fused features, providing a more powerful representation for the final anomaly prediction.

Our ablation study provides strong evidence for the effectiveness of each component in the Dyn-ConvNet model. We find that the synergistic interaction between the local wavelet stream, the global Fourier stream, and the dynamic gating unit is key to achieving superior performance. The wavelet features provide the foundational local context, while the gating mechanism intelligently integrates the multi-scale information. Although the contribution of Fourier features is dataset-dependent, their positive impact on several datasets highlights the value of our hybrid architecture. Overall,

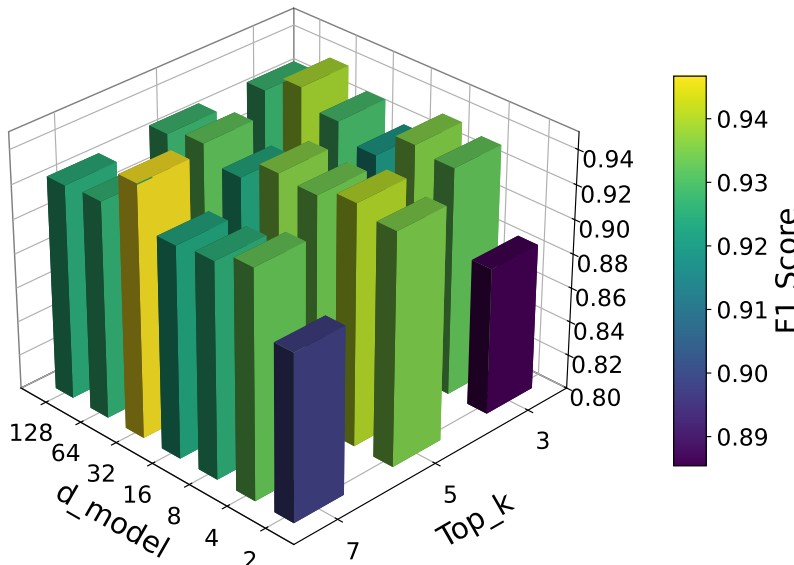

Figure 3: Hyperparameter sensitivity analysis for Dyn-ConvNet on the SWaT dataset. F1-scores across different combinations of $d_{\text{model}}$ and *top-k* illustrate the stability and performance trends of the model.

the full Dyn-ConvNet model demonstrates the most stable and robust performance, validating our multi-component, adaptive fusion approach for multivariate time series anomaly detection.

### 4.6 HYPERPARAMETER SENSITIVITY ANALYSIS

To evaluate the robustness of Dyn-ConvNet and guide its optimal configuration, we conducted a comprehensive hyperparameter sensitivity analysis on the SWaT dataset. Specifically, we examined the influence of two key hyperparameters: the model's hidden dimension ($d_{model}$) and the number of dominant Fourier components (*top-k*). The F1-score was used as the primary performance metric to ensure consistency across experiments.

#### 4.6.1 EXPERIMENTAL DESIGN

We performed a grid search over $d_{\text{model}} \in \{2, 4, 8, 16, 32, 64, 128\}$ and top-$k \in \{3, 5, 7\}$ to evaluate model sensitivity and stability.

#### 4.6.2 RESULTS AND DISCUSSION

Figure 3 summarises the experimental results, presenting the F1-scores for all tested hyperparameter combinations.

Our analysis reveals two important insights: robust performance across a wide range of settings and clear trends linking model capacity with feature richness.

**Impact of $d_{\text{model}}$**   Across all values of *top-k*, Dyn-ConvNet consistently maintains strong performance, demonstrating its robustness. Lower dimensions ($d_{\text{model}} = 2$) exhibit relatively lower F1-scores (0.8854–0.9352), reflecting insufficient capacity to capture the complex patterns in the SWaT dataset. Increasing $d_{\text{model}}$ markedly improves performance, with values between 4 and 32 yielding consistently high F1-scores ($>0.92$). Notably, $d_{\text{model}} = 32$ achieves the highest peak F1-score of 0.9467 for *top-k*=7, highlighting an optimal balance between capacity and generalisation. Beyond $d_{\text{model}} = 32$, performance gains saturate or slightly decline, suggesting diminishing returns and potential overfitting while increasing computational cost.

**Impact of *top-k***   The number of Fourier components influences performance in a dimension-dependent manner. For smaller dimensions ($d_{\text{model}} = 4$), a moderate *top-k*=5 achieves the best performance (0.9400), while for larger dimensions ($d_{\text{model}} = 32$), *top-k*=7 is optimal, enabling the

model to better leverage richer frequency information. This interplay between $d_{\text{model}}$ and *top-k* underscores the importance of coordinated hyperparameter tuning.

Overall, Dyn-ConvNet exhibits remarkable stability across diverse hyperparameter settings, with performance consistently above 0.89. The hyperparameter study confirms that the combination $d_{\text{model}} = 32$, *top-k*=7 offers the best performance (F1=0.9467) on SWaT, and highlights the critical balance between model capacity and feature complexity.

## 5 CONCLUSION AND FUTURE WORK

In this paper, we proposed Dyn-ConvNet, a novel hybrid deep learning architecture for time series anomaly detection. By systematically integrating global periodic features from FFT and local transient features from the Wavelet Transform through an adaptive, deep convolutional backbone, our model successfully addresses the limitations of single-domain approaches. Extensive experiments on five challenging benchmarks demonstrate that Dyn-ConvNet achieves state-of-the-art performance, with an average F1-score improvement of over 1.46% compared to previous methods.For future work, we plan to explore the integration of more advanced wavelet families and apply the Dyn-ConvNet architecture to other time series analysis tasks, such as forecasting and classification.

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

## A    DATASETS

We evaluated Dyn-ConvNet on five widely-used public benchmark datasets for multivariate time series anomaly detection. A detailed description of each dataset is provided below, with a summary of key statistics presented in Table 3.

- **SMD** Su et al. (2019): This dataset, collected from a large internet company, consists of multivariate time series data from numerous server machines. It is characterized by its strong periodic and seasonal patterns, making it an ideal benchmark for evaluating a model's ability to capture global temporal behaviors. The anomalies are diverse and mimic real-world system failures.

- **MSL** Hundman et al. (2018): The MSL dataset comprises telemetry data from the Curiosity rover on Mars. It is a highly complex, multivariate dataset with non-stationary characteristics due to the rover's varying operational modes. Anomalies are often related to sensor faults or unexpected operational events, posing a challenge for models that rely on strict data stationarity.

- **SMAP** Hundman et al. (2018): This NASA dataset contains telemetry data from the SMAP satellite. It is a challenging benchmark due to its diverse mix of anomaly types, including point anomalies (single spikes), contextual anomalies (deviations within a specific context), and collective anomalies (sequences of abnormal data points). This heterogeneity makes it a robust test for a model's generalization capability across different anomaly patterns.

- **SWaT** Mathur & Tippenhauer (2016): The SWaT dataset is generated from a physical water treatment testbed. The data includes both normal operations and deliberate cyber-attack scenarios. It is a key benchmark for evaluating a model's sensitivity to abrupt, transient anomalies that manifest as sudden and significant deviations from normal behavior.

- **PSM** Abdulaal et al. (2021): The PSM dataset consists of aggregated metrics from multiple application servers at eBay. It is notable for its high dimensionality and complex temporal dependencies. Anomalies in this dataset are predominantly collective in nature, representing synchronized events or system-wide issues that affect multiple sensors simultaneously. This makes it a strong test for models capable of handling large-scale, multivariate data.

Table 3: Dataset descriptions. The dataset size is organized in (Train, Validation, Test).

| Tasks | Dataset | Dim | Series Length | Dataset Size |
|---|---|---|---|---|
| | SMD | 38 | 100 | (566724, 141681, 708420) |
| | MSL | 55 | 100 | (44653, 11664, 73729) |
| Anomaly Detection | SMAP | 25 | 100 | (108146, 27037, 427617) |
| | SWaT | 51 | 100 | (396000, 99000, 449919) |
| | PSM | 25 | 100 | (105984, 26497, 87841) |

## B    BASELINES

We selected the following baseline approaches to evaluate Dyn-ConvNet and compare it with state-of-the-art (SOTA) methods on multivariate time series datasets:

- **Transformer** (Vaswani et al., 2017a): The foundational self-attention model that captures dependencies across all time steps, serving as the basis for many modern time series architectures.

- **Reformer** (Kitaev et al., 2020): An efficient variant of the Transformer that addresses the quadratic complexity of self-attention through locality-sensitive hashing (LSH), making it scalable for very long sequences.

- **Informer** (Zhou et al., 2021): Designed to handle long-sequence time series forecasting, Informer uses a ProbSparse self-attention mechanism and a self-attention distilling approach to improve efficiency and performance.

- **Pyraformer** (Liu et al., 2021): This model introduces a pyramid attention mechanism to capture multi-scale dependencies with logarithmic complexity, providing a more efficient way to model long-range temporal patterns.

- **Autoformer** (Wu et al., 2021): A Time Series Transformer that decomposes time series into trend and seasonality components and uses an Auto-Correlation mechanism to find period-based dependencies, enhancing its ability to handle long-term patterns.

- **DLinear** (Zeng et al., 2023): A simple yet highly effective linear model that decomposes time series into trend and remainder components, challenging the notion that complex deep learning models are always necessary for time series forecasting.

- **ETSformer** (Woo et al., 2022): This model integrates the principles of Exponential Smoothing (ETS) with a Transformer architecture, providing a new way to model seasonality and trend in time series.

- **LightTS** (Zhang et al., 2022): An efficient and lightweight Transformer-based model that uses a non-autoregressive strategy to achieve faster inference and lower memory usage for time series forecasting.

- **FEDformer** (Zhou et al., 2022): A frequency-enhanced Transformer that incorporates a mix of Fourier and self-attention mechanisms, aiming to capture both global periodic patterns and local dependencies with high efficiency.

- **TimesNet** (Wu et al., 2022): This model transforms 1D time series into 2D tensors based on multiple periods, allowing it to leverage 2D kernels for multi-scale feature extraction, effectively capturing periodic patterns.

- **KAN-AD** (Zhou et al., 2024): A model that applies Kolmogorov-Arnold Networks (KAN) to the anomaly detection task, providing a novel approach to learning complex functions for time series, which is known for its interpretability and expressiveness.

## C    LLM Usage Instructions

Portions of this paper were polished and proofread with the assistance of a large language model.

