# OpenReview forum: "Dyn-ConvNet: Hybrid Time–Frequency Learning for Time Series Anomaly Detection"
_ICLR.cc/2026/Conference — ICLR 2026 Conference Withdrawn Submission_

### Official Review · Reviewer_JU9d · 2025-10-30

**Soundness:** 2
**Presentation:** 2
**Contribution:** 2
**Rating:** 4
**Confidence:** 5

**Summary:**

The paper introduces a novel hybrid deep learning architecture Dyn-ConvNet for time series anomaly detection that unifies time- and frequency-domain representations. Specifically, Dyn-ConvNet integrates global periodic features from the FFT with localized transient features extracted via the Wavelet Transform. These complementary features are fused through a deep convolutional backbone equipped with dynamic gating mechanisms and residual connections, enabling adaptive weighting between global and local information depending on the anomaly context. Comprehensive experiments on five multivariate benchmark datasets (SMD, MSL, SMAP, SWaT, and PSM) demonstrate that Dyn-ConvNet achieves superior performance.

**Strengths:**

S1. A major strength of Dyn-ConvNet lies in its adaptive gating mechanism, which dynamically balances the contributions from the time and frequency branches.

S2. Extensive experiments across five benchmark datasets show that Dyn-ConvNet consistently outperforms compared models.

S3. The paper includes detailed ablation experiments that isolate the effects of each core module—Fourier branch, wavelet branch, gating mechanism, and core convolution block. These studies convincingly demonstrate the complementary benefits of the time-frequency fusion strategy and validate the necessity of adaptive gating for robust detection performance.

**Weaknesses:**

W1. While Dyn-ConvNet presents a well-structured hybrid framework, its core idea of combining time- and frequency-domain representations is not entirely novel. Recent works such as TFMAE [1] and TFAD [2] have already explored similar hybrid paradigms that integrate temporal and spectral features for anomaly detection. Although the paper introduces a gating fusion mechanism, this component alone may not represent a sufficiently groundbreaking methodological advance to justify the claim of being “the first systematic integration” of time–frequency features. As such, the incremental novelty of the approach is a key limitation.

W2. The experimental evaluation, while extensive, omits several recent and highly competitive baselines in time series anomaly detection, such as DADA [3] and DCdetector [4]. Without including these baselines, the paper’s empirical comparisons risk overstating the performance advantage of the proposed model.

W3. Although the paper provides quantitative results and ablation studies, it lacks qualitative case studies or visual analyses demonstrating how Dyn-ConvNet handles different anomaly types (e.g., point, contextual, and collective anomalies).

W4. The paper does not include experiments on large-scale datasets or report metrics related to computational efficiency. Since hybrid FFT–wavelet models can be computationally expensive, it is important to evaluate scalability on longer sequences or higher-dimensional data.

[1] Temporal-Frequency Masked Autoencoders for Time Series Anomaly Detection

[2] TFAD: A Decomposition Time Series Anomaly Detection Architecture with Time-Frequency Analysis

[3] Towards a General Time Series Anomaly Detector with Adaptive Bottlenecks and Dual Adversarial Decoders

[4] DCdetector: Dual Attention Contrastive Representation Learning for Time Series Anomaly Detection

**Questions:**

See Weaknesses.

---

### Official Review · Reviewer_YCAh · 2025-10-31

**Soundness:** 2
**Presentation:** 3
**Contribution:** 1
**Rating:** 2
**Confidence:** 3

**Summary:**

This paper proposes Dyn-CovNet, a hybrid network which integrates both FFT and Wavelet transform. The FFT can extract long-range periodic features and Wavelet transform can extract local features. These features are fused through a deep convolution backbone supported by gating mechanisms and residual connections. Experiments on five multivariate time series datasets demonstrate the good performance of the proposed Dyn-ConvNet.

Specifically, Dyn-ConvNet uses a dual-branch architecture: 1) a Fourier stream extracts dominant frequency components; and 2) a Wavelet stream captures fine-grained temporal changes via multi-scale convolutions. These features are fused through a gating mechanism guided by Fourier features. A residual convolution block further refines the fused representation for reconstruction-based anomaly scoring.

**Strengths:**

1. The network proposed is rather straightforward and the paper is easy to read.
2. The proposed network considers global periodic features using FFT and local features using Wavelet transform.

**Weaknesses:**

1.	Flawed experiments. The paper addresses a time series anomaly detection task. In experiments, all compared baselines are forecasting algorithms. It is strongly recommended to compare these SOTA time series anomaly detection algorithm, which can be found in [R1].

References:
R1.	Zahra Zamanzadeh Darban, et al. Deep Learning for Time Series Anomaly Detection: A Survey.

**Questions:**

1.	What are B and L in Line 148?
2.	The FFT provides the global frequency information and the wavelet transform provides the time-frequency information. By perform aggregation over the time, we can also extract global frequency information from the results produced by wavelet transform. Thus, I would suggest the authors explore using wavelet transform alone without FFT for time series anomaly detection.

---

### Official Review · Reviewer_QhQZ · 2025-11-01

**Soundness:** 2
**Presentation:** 3
**Contribution:** 2
**Rating:** 2
**Confidence:** 4

**Summary:**

This work studies a hybrid time-frequency network for time series anomaly detection. The time domain employs multi-scale convolution, while the frequency domain utilizes the Fourier Transform. A gating network is then used for integration, with anomalies being detected based on model reconstruction.

**Strengths:**

- The paper is well-presented.
- Basic experimental comparisons and ablation studies are included.

**Weaknesses:**

- However, despite the presence of the experiments, there are no significant improvements or in-depth analyses observed. First, the results do not show consistent improvements. Second, the integration of the time and frequency domains in these two components lacks significant contribution, as existing works have already explored joint representation learning or anomaly detection in the time-frequency domain.

- The terminology used is not entirely accurate. The authors refer to the Wavelet Transform, but they actually use multi-scale convolution. This is misleading since the authors are not employing the standard wavelet transform from signal processing. Additionally, when the convolution kernel size equals the length of the input sequence, it effectively becomes equivalent to global Fourier features. Therefore, the necessity of merging these two branches requires further discussion, such as identifying what types of anomalies each branch helps address. I did not find any in-depth analysis on this in the experiment part.

- Moreover, the wavelet transform you mentioned is also a method for frequency modeling, right? From this perspective, the emphasis on "time-frequency" in the title is not rigorous.

- The review of related works is insufficient. The comparison methods mainly focus on time series prediction models, such as DLinear and Transformer variants, lacking a thorough discussion of various anomaly detection works. Specifically, the authors state in the abstract that “Existing methods typically operate in a single domain—either time or frequency.” This is unreasonable, as there is actually a substantial amount of work that utilizes time-frequency fusion or multi-scale analysis methods for time series anomaly detection.

- Figure 3 suggests that the model's sensitivity is dependent on specific tuning operations.

- The gating mechanism simply concatenates the representations from the two branches and employs a linear projection for gating selection. This design seems somewhat arbitrary, as the feature learning capability of this gating network is quite weak.

**Questions:**

See Weaknesses.

---

### Official Review · Reviewer_wc6v · 2025-11-01

**Soundness:** 2
**Presentation:** 2
**Contribution:** 1
**Rating:** 2
**Confidence:** 3

**Summary:**

The paper proposes Dyn-ConvNet, a hybrid deep learning architecture for multivariate time series anomaly detection (TSAD). The central premise is that existing models are limited by operating in a single domain (either time or frequency). To address this, Dyn-ConvNet introduces a dual-stream architecture: (1) an "Adaptive Fourier Feature Stream" using FFT to capture global periodic patterns and (2) a "Local Wavelet Feature Stream"  (implemented via multi-scale dynamic convolution) to capture local, transient patterns. These streams are combined using an adaptive gating mechanism and processed by a core convolutional block for signal reconstruction. The model is evaluated on five public benchmarks, where it claims to achieve state-of-the-art F1-scores.

**Strengths:**

1. The paper addresses the important and challenging task of TSAD, and the motivation to handle both global and local anomalies simultaneously is well-founded.
2. The authors conduct experiments on five widely-used multivariate benchmarks (SMD, MSL, SMAP, SWaT, PSM) and include a detailed ablation study .

**Weaknesses:**

1. Inconsistent Empirical Support for the Core Premise: The paper's primary hypothesis is that the integration of time and frequency domains is key to its superior performance. However, the ablation study (Table 2) does not consistently support this conclusion.
When comparing the "No-FFT" variant (time-domain only) to the "No-Gating" variant (a simple hybrid concatenation), the "No-FFT" model achieves a significantly higher F1-score on SMAP (0.8358 vs. 0.6869) and also performs better on SWaT and MSL.
This result on a majority (3 out of 5) of the datasets strongly suggests that the Fourier stream is not always complementary and, in many cases, may actually be detrimental to performance . This finding undermines the central justification for the proposed hybrid architecture.

2. The role of the adaptive gating mechanism becomes unclear . Given that the FFT stream appears to harm performance on several datasets, the "Full Model's" high scores might be achieved simply by the gate learning to suppress or ignore the FFT input on those datasets. If this is the case, the model is effectively a complex, gated CNN, and the hybrid-domain premise adds unnecessary complexity. The paper lacks a sufficient analysis (e.g., visualizing the gate weights) to disambiguate this.

**Questions:**

1. The ablation study (Table 2) shows that removing the FFT stream ("No-FFT") results in superior F1-scores on 3 of the 5 datasets (SMAP, SWaT, MSL) compared to the "No-Gating" baseline . This proves the Fourier stream is often detrimental. How do the authors reconcile this fact with the paper's central thesis that integrating time and frequency domains is the key to improved performance ?

2. Can the authors please clarify if a mathematical Wavelet Transform (e.g., CWT or DWT) is ever computed? The text repeatedly uses the term "Wavelet" , but the implementation is described as "multi-scale convolutions (kernel sizes 3 and 5)" and "adaptive dynamic convolution". If it is the latter, the authors must correct this misleading terminology throughout the paper.

---

### Note · Authors · 2025-11-12

I have read and agree with the venue's withdrawal policy on behalf of myself and my co-authors.